# Association between Feelings of Trust and Security with Subjective Health among Mexican Migrants in the New York City Area

**DOI:** 10.3390/ijerph20042981

**Published:** 2023-02-08

**Authors:** Pablo Gaitán-Rossi, Mireya Vilar-Compte, Isabel Ferré-Eguiluz, Luis Ortiz, Erika Garcia

**Affiliations:** 1Research Center for Equitable Development EQUIDE, Universidad Iberoamericana, Prolongación Paseo de la Reforma 880, Lomas de Santa Fe, Mexico City 01219, Mexico; 2Department of Public Health, Montclair State University, Montclair, NJ 07043, USA

**Keywords:** migration, subjective health status, social context, trust, security

## Abstract

The size of the foreign-born population living in the United States makes migrants’ health a substantive policy issue. The health status of Mexican immigrants might be affected by the level of social capital and the social context, including the rhetoric around immigration. We hypothesize that a diminished perception of trust and safety in the community has a negative impact on self-reported health. In a cross-sectional study, we conducted a survey among 266 Mexican Immigrants in the New York City Area who used the Mexican Consulate between May and June 2019 for regular services provided to documented and undocumented immigrants. A univariate and bivariate descriptive analysis by trust and security items first shows the diversity of the Mexican population living in the US and the conditions of vulnerability. Then, logistic regression models estimate the association between trust and security items with self-reported health status. Results show that safety is consistently associated with good self-rated health, especially when rating the neighborhood, and trust showed mixed results, more reliant to the way it is operationalized. The study illustrates a pathway by which perceptions of the social context are associated with migrants’ health.

## 1. Introduction

The United States (US) has long been a recipient of immigrants from a wide range of countries, of which Mexico is an important one. Such migratory process affects migrants’ lives as well as the recipient communities. Mexican immigration to the US has historically been a significant phenomenon. From 2000 to 2017, the population of Mexican-born migrants living in the US grew from 8.7 to 11.2 million; likewise, the Hispanic population of Mexican origin living in the US (both migrants and those who trace their family ancestry to Mexico) grew from 20.9 to 36.6 million (a 76% increase) in the same period, accounting for 62% of the US Hispanic population [1]. Mexican immigrants alone account for 25% of all US immigrants [2]. By itself, Mexican immigration is a substantive policy issue due to its magnitude and its political visibility. 

The characteristics of Mexican immigrants to the US have changed over time. Traditional immigration (pre-1980s) was mainly comprised of male adults and youths who came mostly to work in seasonal and manufacturing-related jobs. Most migrants came from specific areas in Mexico and settled in a limited number of counties in California, Texas, Arizona, and Illinois [3]. In contrast, new flows of Mexican immigrants (post-1980s) come from diverse places in Mexico and settle in varied locations in the US. Furthermore, according to the Department of Homeland Security’s 2015 estimates, 55% of the 12 million undocumented immigrants were from Mexico [4]. Mexican immigrants to the New York City Area (NYCA) are an example of this new flow of immigrants. They comprise the foreign-born group with the third highest population growth in New York City between 2000 and 2015, with a 47% growth in this period, only behind China and Bangladesh [5]. 

Health is one of the dimensions affected by immigration [6]. There is a dynamic relationship between attitudes, behaviors, and access to healthcare services with health outcomes among Mexican immigrants. Previous studies have reported that Mexican immigrants have some favorable health behaviors and health outcomes as compared to native-born. However, it has also been found that as Mexican immigrants acculturate, this “health advantage” is eroded [7]. While acculturation has been assumed to have a prominent role in determining health behaviors and disease risk through the effect it has on behaviors, norms, and attitudes [8,9,10], it has been less documented how the difficulties faced by undocumented immigrants change when the national political mood towards immigration is modified by the social context. Such changes are likely to affect access to healthcare services, as well as to generate stress with potential negative effects on health outcomes [11,12].

One of the challenges faced by Mexican immigrants is having access to healthcare services. The implementation of the Affordable Care Act (ACA) during President Obama’s administration excluded undocumented immigrants and ratified waiting periods for Medicaid-eligible immigrants. However, it supported additional funding for over five years for Federally Qualified Health Centers (FQHCs), which provide medical services for uninsured populations. Even though an extension seemed unlikely, in 2017, the House of Representatives approved a two-year extension until 2019. Despite this extension, however, the sustainability of FQHCs remains a permanent concern, as uninsured immigrants rely heavily on FQHCs as well as on other safety-net providers to meet their healthcare needs [13].

Another challenge that Mexican immigrants deal with is healthcare utilization. According to the literature, Mexican immigrants, especially those who are undocumented, are less likely to access and use healthcare services compared to US-born Mexican Americans, other Latinos, and non-Latino whites [14,15,16]. Socio-economic status, language barriers, lack of transportation, and lack of work authorization are amongst the most common barriers that Mexican immigrants, along with other immigrants in the US, face to access and use healthcare services [17]. However, fear of deportation and discrimination are also important barriers to healthcare access and healthcare utilization, especially among undocumented immigrants [18]. 

Anti-immigrant policies and rhetoric, such as those observed during President Trump’s administration, may be further creating fear-based barriers to healthcare that affect not only undocumented immigrants, but legal permanent residents as well, regardless of their health insurance coverage status [19]. Fear regarding these policies and narratives could be discouraging immigrants from using healthcare. Proposed rules on the definition of “public charge” resulted in lower utilization rates of health, nutrition, and other public benefit programs among immigrant families, including those families that are not directly affected by the new public charge rule [20,21]. These public charge rules would make it harder for poor immigrants who have received public benefits to obtain US citizenship and will discourage them, and their US-born children, from enrolling in subsidized healthcare programs, such as Medicaid, the Supplemental Nutrition Assistance Program (SNAP), the Medicare Part D Low-Income Subsidy Program, and other housing programs. These changes could result in substantial negative impacts on the health of Mexican immigrants (e.g., considering that some families might avoid enrollment or disenroll themselves and their children from public programs, families could forgo critical medical care) for fear of decreasing their chances of gaining permanent residency status [21,22].

In addition, evidence shows that changes in immigration enforcement actions have resulted in a greater fear of deportation (even among naturalized citizens and legal permanent residents), thus generating an atmosphere of anxiety, stress, and depression that may ultimately result in the worsening of health outcomes and well-being of immigrant families [23,24,25]. Prior work documents that a large proportion of Hispanics (both citizens and noncitizens) feel unsafe living in the US because of the anti-immigrant narratives, which have fostered fear and mistrust [26]. This is likely to affect the perceptions about the communities and contexts where immigrants live [27], including aspects such as neighborhood safety and trust. This is worrisome as prior research suggests that such neighborhood and community perceptions can have implications in subjective [28,29,30] and objectives measures of health [31].

The objective of this study was to assess how the self-reported health status is associated with perceptions of trust and safety among Mexican immigrants living within the NYCA during President Trump’s administration, when anti-immigrant narratives where salient. 

## 2. Methods

### 2.1. Study Design and Sample

The cross-sectional study was conducted among Mexican immigrants who mostly resided within the NYCA. Mexican immigrants aged 18 years and older, who identified themselves as being residents of the US, were asked to respond to a survey at the main office of the Consulate General of Mexico in New York City in Manhattan or at the Consulate’s mobile offices (“*Consulados móviles*”), which travel to communities outside of Manhattan. Both offices serve Mexican immigrants living within the tri-state area (i.e., New York, Connecticut, and thirteen counties in New Jersey) and provide services to both documented and undocumented immigrants. Services include the renewal of Mexican passports, the issuance of consular IDs, legal counselling, and health services (provided by the “*Ventanillas de Salud*”), amongst others. The advantage of conducting the study in these two settings is linked to three elements. First, many Mexican immigrants residing within the tri-state area, regardless of immigration status, visit the Consulate’s offices to obtain a consular ID, which is a valid identity document in the US. Second, appointments at the Consulate’s offices are scheduled by phone on a random basis by Consulate staff. Third, Mexican immigrants are more open to talk and to participate within the premises of the Consulate, where they do not fear about their migratory status.

Immigrants completing paperwork at the Consulate’s offices were approached individually by previously trained researchers from Universidad Iberoamericana and invited to participate in the study. The research team informed potential participants about the study details and asked them whether they would like to respond to a face-to-face survey, which took approximately 20 min to complete. All immigrants who agreed to participate in the study and who met the eligibility criteria were then asked to sign an informed consent form. Then, the research team administered the survey to participants by using electronic tablets that had been loaded with the Magpi mobile data collection software. The survey instrument had been previously piloted and was available both in Spanish and in English. Convenience sampling was carried out at both the Consulate’s and the Consulate’s mobile offices’ waiting areas and at the area near the “*Ventanilla de Salud*”, where two researchers scouted for participants at all times to cover all waiting areas. Due to the dynamic flow of people in the waiting areas (i.e., the same people moved between sections of the waiting areas at a fast pace during their visit), the research team was unable to conduct a systematic sampling procedure; however, most of the people who were completing paperwork at the Consulate’s offices were asked, at some point, whether they were interested in participating in the study. In addition, although our sampling strategy did not involve a systematic approach on gender, the research team approached people, regardless of gender, and asked them whether they were interested in participating in the study. 

Data collection took place between 1 May and 12 June 2019, and survey participants were not compensated for participating in the study. The estimated participation rate was 76%. The research team administered the survey to a total of 266 participants. However, 10 of them could not complete the survey as their paperwork at the Consulate’s offices concluded before getting to the end of the survey.

The research protocol was reviewed and approved by the Universidad Iberoamericana Ethics Committee for Research in Mexico City.

### 2.2. Study Variables

#### 2.2.1. Subjective Health Status

Participants self-rated their health as excellent, very good, good, fair, or bad. For purposes of the analysis, we generated a dummy categorical variable (i.e., excellent, very good, good health (1), and fair or bad health (0)). Self-reported health status is a subjective health measure that is commonly used in population-based studies [32]. Furthermore, prior studies highlight that poor self-rated health can be a predictor of mortality [33], as well as pathological changes prior to disease diagnosis [34]. 

#### 2.2.2. Measures of Subjective Perceptions of Trust and Safety 

We operationalized the measures of trust and safety in different ways to disentangle their specific effects on self-reported health. First, we assessed items separately: *trust* was measured with the item “I can trust most people in my community”, *help* with the item “I can get help from my neighbors whenever I need it”, *individual safety* with the item “I feel safe when I walk alone at night in my community”, and *neighborhood safety* with a rating of how secure they consider their neighborhood. Items used 4 Likert-type response options and were scored from low-to-high. To separate self-perceptions of trust and safety from individual perceptions about the neighborhood, we then added the first three items to create a trust score according to what has already been used in prior studies [29,35] as they have been found to be associated with adverse health outcomes and poorer use of healthcare services among vulnerable populations [36,37,38]. Lastly, we added the four items because preparatory analyses indicated that the four items have moderate-to-high correlations (0.3–0.6). Therefore, we used two composite measures, the three-item score of trust, ranging from 0 to 3, and the four-item composite measure of trust and safety (i.e., individual and neighborhood), ranging from 0 to 4; the higher the score, the greater the individual’s trust and sense of safety. The three-item and four-item measures had moderate reliability, with a Cronbach’s alpha of 0.61 and 0.65, respectively.

#### 2.2.3. Sociodemographic Characteristics

Sociodemographic characteristics collected during the survey included age, gender, employment status (i.e., not working or looking for a job vs. working), and household composition (i.e., with and without children). A proxy for acculturation was the length of residence, a continuous variable of the number of years lived in the US. 

Comorbidities were measured using information on seven self-reported health issues—diabetes, hypertension, heart disease, asthma or bronchitis, cancer, depression, and arthritis. These problems represent top causes of morbidity among the study population [39]. Survey respondents were asked if a physician or another medical professional had told them that they had one of these conditions. Dummy variables were generated for each condition, then a summative variable showed the total number of comorbidities (i.e., from 0 to 7), and lastly, we created a dummy variable (i.e., having at least one comorbidity vs. not). A large body of empirical research highlights the effect of health insurance in accessing healthcare [40,41]; hence, participants were asked if they had a valid health insurance plan in the US (operationalized as a dummy variable). Furthermore, participants were asked to indicate how frequently they had received non-urgent care when they needed it, and this led to a categorical variable (never or sometimes/usually vs. always). These are common proxy measures generally used when studying vulnerable populations [42]. 

To ascertain the immigrant’s community of residence, respondents were asked to provide their zip code. With this information, we used geographic referencing and clustered Mexican American residents throughout the NYCA.

### 2.3. Statistical Analysis

An initial analysis described the characteristics of the survey population. A subsequent bivariate analysis compared the characteristics by self-reported health status.

Then, we estimated six logistic regression models to assess the association of multiple operationalizations of trust and safety on self-rated health, adjusted by covariates (i.e., age, gender, household composition, employment status, length of residence, frequency of receiving medical care when needed). The first model estimated the association using the four items of trust and safety operationalized as independent dummy variables. In the second model, only the *neighborhood safety* item was included to assess its distinct contribution. The third model estimated the association of the three-item score of trust, and in the fourth model the three-item score of trust was kept, and the *neighborhood safety* item was added as an independent dummy. The fifth model estimated their two-way interaction (three-item score of trust and *neighborhood safety*). Lastly, the sixth model estimated the association of the four-item score of trust and safety. Figure 1 summarizes this analytical approach.

This strategy allowed us to examine distinct associations and address different identification issues linked to trust and safety perceptions. Forest plots were used to communicate the size and significance of regression coefficients, and the full table with confidence intervals is provided in the Appendix A. All statistical analyses were performed with R and the plot in the *sjPlot* package (version 2.8.12) [43]. In addition, to complement the descriptive analyses, we geo-referenced the participants’ residential zip codes with QGIS 3.4 [44].

In the Appendix A, we show the full identification process of the logistic regressions: first with univariate regression analysis for the main independent variables, then with the full model, and lastly with significant variables only. These sensitivity analyses support the main findings. 

## 3. Results

As shown in Table 1, survey respondents were 39 years of age on average, and a slightly higher proportion were females (56.7%). Almost two thirds of the households had children (65.5%) and reported to be working (75.6%) at the time of the survey. On average, participants had lived in the US for 17 years. Half of the sample self-reported a fair or bad health status (51%), which is surprising considering their average age. Nonetheless, 42% were diagnosed with at least one comorbidity. In terms of access to healthcare, only about one-third of the sample (37.6%) reported having a valid health insurance plan in the US and 56.4% acknowledged they did not receive care when needed. In terms of the respondents’ communities of residence, Figure 2 portrays the areas where participants currently reside in the US. Map 1 shows that the majority live in the state of New York and Map 2 zooms into the New York City Area, highlighting that most of them live in Brooklyn, Queens, and to a lesser degree in the Bronx. In Brooklyn and Queens, there seem to be areas with more clustered Mexican immigrant communities.

Perceived trust and safety measures showed that 39.3% of the participants distrust people in their communities, 25.8% do not believe they would get help from their neighbors when needed, and 18.1% perceived their communities as an unsafe place to walk alone at night (Table 1). Moreover, 13.7% of the participants stated that the neighborhood where they lived was insecure. The three-item composite score of trust had an average of 2.17 and the four-item composite score of trust and safety has an average of 3.04, indicating high levels of trust and security in the sample.

Table 2 summarizes the bivariate analysis by self-reported health status. Sociodemographic characteristics did not significantly vary. As expected, those with comorbidities reported lower self-reported health levels, but there were no differences by health insurance and frequency of healthcare when needed. The items linked to trust, and the trust composite score, did not show statistical differences by self-reported health. However, the items linked to individual safety and the perceptions of safety in the neighborhood indicate that those who report poorer health also tended to feel unsafe, and this association remained in the trust and safety composite score (Table 2).

Figure 3 presents the results of the six regression models, and the lower part shows the coefficients of the independent variables. The first model shows the four items separately when adjusting for covariates. The strongest effect was with the positive association between neighborhood safety and self-reported health (OR 3.52). Notably, the trust items showed contradictory effects depending on the specific operationalization. While trusting people in the community had a negative effect, getting help from neighbors had a positive effect. Model 2 examined the single association of neighborhood safety on self-reported health and found that the size of the effect remained almost unchanged (OR 3.59) when compared to the first model.

The three-item trust score used in model 3 showed a positive but non-significant association with self-rated health, most likely due to the item on trust in the community. The inclusion of the neighborhood item in model 4 confirmed its importance, although the effect was slightly lower (OR 3.38). Model 5 tested the interaction between the trust score and the neighborhood safety item and found a non-significant effect of the interaction, but the size of the neighborhood item increased (OR 5.16), suggesting the importance of the perceived insecurity in the neighborhood amongst those who trust their neighbors. Lastly, model 6 tested the effect of the four-item trust and safety score on subjective health. As hypothesized, the main effect was positive and significant (OR 1.26). The most important confounders in all models were comorbidities and the frequency of receiving healthcare when needed, but the length of time residing in the US and having valid health insurance in the US were not as salient in the models as expected.

## 4. Discussion

The Mexican population living in the NYCA is diverse and it probably confounds several profiles. For instance, the Consulate General of Mexico in New York equally serves acculturated individuals that have been living for a long time in the US and less-acculturated individuals who arrived around a year ago. Likewise, they vary in working status and live in different states, counties, and neighborhoods, where they must learn to navigate different health-system regulations.

The survey confirmed that the Mexican population in the NYCA faces important conditions of vulnerability regarding health. They reported remarkably low levels of self-reported health for their age and above-average levels of diagnosed cardiovascular disease (i.e., while 12.45% of individuals in our sample reported a diagnosis of diabetes, the prevalence for Hispanics living in New York was 11% in 2019 and 10.3% for all ages of the Mexican population) [45]. This is aggravated by the fact that most of the sample population were uninsured (62%) and 56% sometimes or always eschewed medical care when needed.

These conditions of vulnerability could be worse for those who do not rely on informal networks for help. The study revealed that 39% of the participants do not trust people in their community and 26% do not believe they would receive help from their neighbors if needed. This is worrisome because social capital is a key driver to migrate and contrasts with the Hispanic cultural values and migratory enclaves where people feel safe [46].

The main finding of the study was the statistical association between the perception of neighborhood insecurity and lower self-reported health status, regardless of the length of residence in the US and having a valid insurance plan in the US. The relevance of the finding is that the measured perception of an insecure environment might correspond to the political context against Mexican immigrants. In this sense, stress is a possible direct pathway explaining the association between safety and self-reported health. Fear of deportation and discrimination might translate into additional anxiety, affecting health outcomes and the perception of health status [47,48]. These effects were present in a “sanctuary regio” such as the NYCA and are probably higher in states with a harsher anti-immigrant narrative. A comparison of these associations with a similar population but in different political climates (either in different periods or different states) would offer a stronger basis to this important finding.

Notably, *trust* showed mixed results on self-rated health because it was more susceptible to the way it is operationalized. While getting help when needed was consistently associated with self-reported health, the more general item on trusting the community had an association in the opposite direction. The *trust item* was higher than the other three items in the univariate analyses and did not show statistical differences with subjective health in the bivariate analyses, suggesting that this item might be capturing additional unobserved topics. Therefore, while the summative scores run in the expected direction and are oftentimes significant, the effect size is probably lower due to the statistical noise coming from the *trust* item. Nonetheless, the correlations between trust and safety suggest that these are two intertwined variables related to self-reported health status. Further studies should attempt to use more complex measures and larger samples to continue disentangling their effects in the migrant population. Furthermore, qualitative evidence could help in understanding, with more granularity, this association and to describe the new strategies Mexican immigrants in the NYCA are following to receive the needed healthcare.

A strength of the current research is that by conducting the data collection at the premises of the Consulate and by Mexican interviewers, the participants were more willing to participate and to share their responses with openness. Similarly, the Consulate allowed capturing a diverse sample of Mexican Americans in terms of acculturation, working status, health status, age, and family structure, amongst others. The study had some limitations, including the use of self-reported diagnosed comorbidities and the small sample size. The bias introduced by self-reported diagnosed comorbidities could have resulted in lower prevalence. However, this is common in population studies and has been widely used [49]. Furthermore, the small sample size employed in the study could also raise concerns. However, we believe that finding statistically significant associations between the trust and safety and self-reported health status highlights the strength of the outcomes. In addition, since the sample refers to a very specific population (i.e., Mexican immigrants in the NYCA), this may compromise the external validity of the study. Future studies should target populations in other sampling sites and geographic areas.

From a policy perspective, this study contributes to highlighting the need for community building activities and participatory-based neighborhood-level interventions in areas in which there are large clusters of Mexican immigrants, such as those shown in the maps, which might be fundamentals to improve health outcomes. The Mexican Consulate, together with community-level organizations, and the local governments can be fundamental actors. It might be relevant to conduct network map analyses to assess who might be other relevant actors in designing and implementing such interventions.

## 5. Conclusions

The study evidenced the deleterious health effects of a diminished sense of trust and safety, such as the ones Mexican immigrants in the NYCA face during a period of anti-immigrant discourse. Even if the study used a small sample size, collecting a survey in a safe space, such as the Consulate, has offered unique data about a hard-to-reach but relevant population. While the study needs to be replicated in other Consulates and States, our results highlighted a singular pathway by which the health advantage of migrants erodes once they are settled into a new country in which anti-immigrant narratives are common ground. Moreover, it provided further evidence of why immigration status is a risk factor for poor health outcomes in the US. The findings underscore the health effects of living in a hostile environment and serve as a cautionary tale, showing how political threats harm vulnerable populations.

## Figures and Tables

**Figure 1 ijerph-20-02981-f001:**
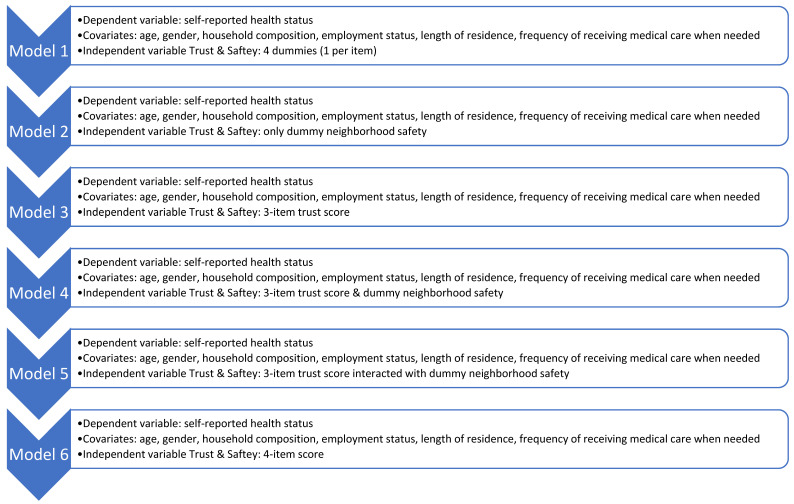
Six-step analytical strategy to assess the association between self-reported health status with trust and safety.

**Figure 2 ijerph-20-02981-f002:**
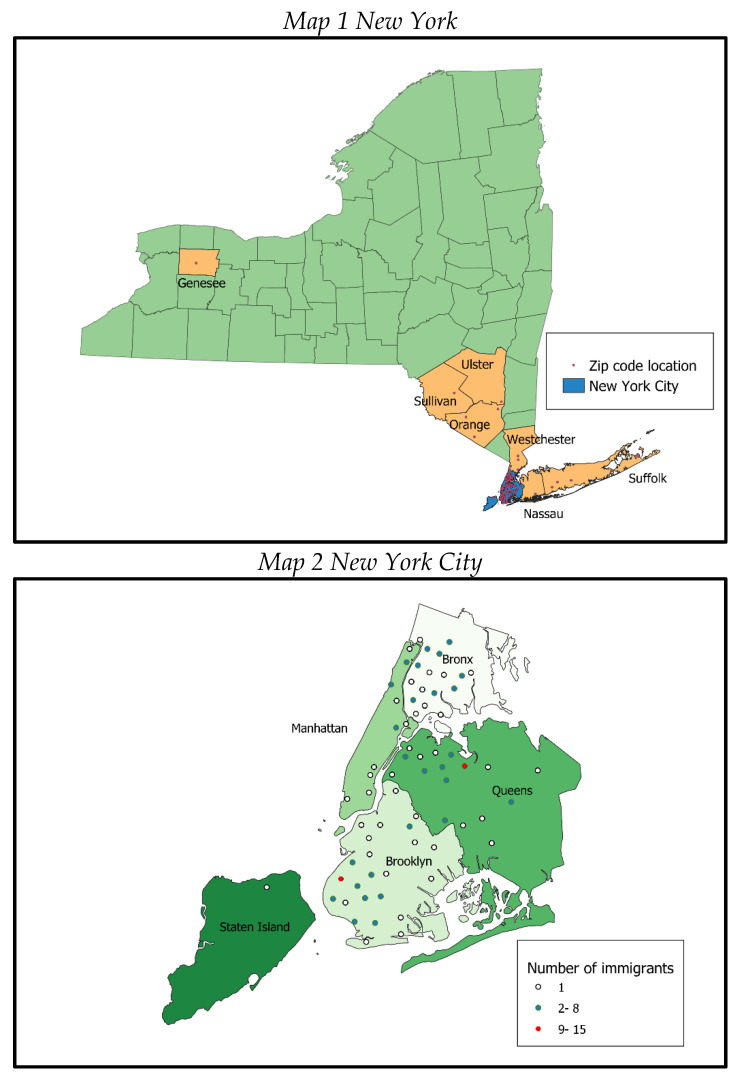
Maps portraying immigrants’ current places of residence. Colors indicate different boroughs of New York.

**Figure 3 ijerph-20-02981-f003:**
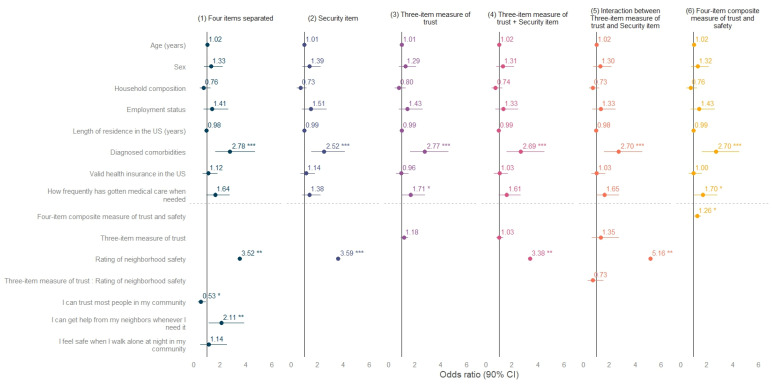
Models on self-reported health by measure of trust and safety. Forest plot with six models, each term of the models is presented on the left axis. The dots and lines illustrate the odds ratios and the confidence intervals (90%), respectively. * *p* < 0.1, ** *p* < 0.05, *** *p* < 0.01.

**Table 1 ijerph-20-02981-t001:** Descriptive statistics of the study sample.

Total (N = 266)		
Variable	% (N)	*p*-Value
*Age*, years (mean, SD)	39.14 (10.78)	
*Sex* (%, n)		0.030 *
Male	43.30% (113)	
Female	56.70% (148)	
Household composition (%, n)		<0.000 *
With children	65.5% (169)	
Without children	34.5% (89)	
*Employment status* (%, n)		<0.000 *
Not working	24.43% (64)	
Working	75.57% (198)	
*Length of residence in the US* (mean, SD)	17.33 (8.18)	
*Diagnosed comorbidities*		0.010 *
One or more diagnosed comorbidities	42.11% (112)	
Without diagnosed comorbidities	57.89% (154)	
*Has valid health insurance in the US* (%, n)	37.6% (97)	
*How frequently has received medical care when needed* (%, n)		0.0403 *
Never or Sometimes	56.37% (146)	
Usually or Always	43.63% (113)	
*Self-reported health status* (%, n)		0.753
Fair or bad	50.99% (129)	
Excellent, very good, and good	49.01% (124)	
Subjective measures of trust and safety		
*I can trust most people in my community* (%, n)		0.001 *
Disagree or Strongly disagree	39.29% (99)	
Agree or Strongly agree	60.71% (153)	
*I can get help from my neighbors whenever I need it* (%, n)		<0.000 *
Disagree or Strongly disagree	25.82% (63)	
Agree or Strongly agree	74.18% (181)	
*I feel safe when I walk alone at night in my community* (%, n)		<0.000 *
Disagree or Strongly disagree	18.11% (46)	
Agree or Strongly agree	81.89% (208)	
*Rating of neighborhood safety* (%, n)		<0.000 *
Insecure or Very insecure	13.73% (35)	
Secure or Very secure	86.27% (220)	
*Three-item measure of trust* (mean, SD)	2.17 (0.99)	
*Four-item composite measure of trust and safety* (mean, SD)	3.04 (1.16)	

* Statistical significance at 95%.

**Table 2 ijerph-20-02981-t002:** Bivariate associations by self-reported health status.

Variable	Self-Reported Health Status (%, n)		*p*-Value
	Poor Health (n = 129)	Good Health (n = 124)	
Age, years (mean ± SD)	40.0 ± 11.2	38.7 ± 10.3	0.348
Sex (%, n)			0.429
Male	46.9% (60)	41.1% (51)	
Female	53.1% (68)	58.9% (73)	
Household composition (%, n)			0.253
With children	61.7% (79)	69.4% (84)	
Without children	38.3% (49)	30.6% (37)	
Employment status (%, n)			0.196
Not working	27.9% (36)	20.2% (25)	
Working	72.1% (93)	79.8% (99)	
Length of residence in the US (mean, SD)	18.0 ± 7.6	16.9 ± 8.8	0.273
Diagnosed comorbidities			<0.001 *
One or more diagnosed comorbidities	55% (71)	32.3% (40)	
Without diagnosed comorbidities	45% (58)	67.7% (84)	
Health insurance (%, n)			0.523
Does not have a valid health insurance in the US	64.1% (82)	59.3% (73)	
Has valid health insurance in the US	35.9% (46)	40.7% (50)	
How frequently has received medical care when needed (%, n)			0.465
Never or Sometimes	58.6% (75)	53.2% (66)	
Usually or Always	41.4% (53)	46.8% (58)	
Subjective measures of trust and safety			
I can trust most people in my community (%, n)			0.961
Disagree or Strongly disagree	39.4% (50)	39.7% (48)	
Agree or Strongly agree	60.6% (77)	60.3% (73)	
I can get help from my neighbors whenever I need it (%, n)			0.099
Disagree or Strongly disagree	30.6% (38)	20.5% (24)	
Agree or Strongly agree	69.4% (86)	79.5% (93)	
I feel safe when I walk alone at night in my community (%, n)			0.018 *
Disagree or Strongly disagree	24.4% (31)	12.1% (15)	
Agree or Strongly agree	75.6% (96)	87.9% (109)	
Rating of neighborhood safety (%, n)			0.002 *
Insecure or Very insecure	21.1% (27)	6.5% (8)	
Secure or Very secure	78.9% (101)	93.5% (115)	
Three-item measure of trust (mean, SD)	2.1 ± 1.1	2.3 ± 0.9	0.123
Four-item composite measure of trust and safety (mean, SD)	2.9 ± 1.3	3.2 ± 1.0	0.019 *

Age, length of residence in the US, and *p*-values are from the t-test, while the other comparisons relied on a chi-square test. * Statistical significance at 95%.

## Data Availability

Data is available upon request.

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
