# Peer review of "Association between Feelings of Trust and Security with Subjective Health among Mexican Migrants in the New York City Area"

_ijerph, 2023, doi:10.3390/ijerph20042981_

Round 1
Reviewer 1 Report
An interesting topic, but the paper is based on weak data, unfortunately; that is, the sample is non-random and small. Perhaps this study can serve as the groundwork for a more extensive research using scientific sampling with a substantially larger sample. This might be pointed out in the conclusion.
Some aspects of the presentation can be improved.
1. Surprisingly, a substantively important variable was not stratified in the analysis. Why not stratify on "documented" vs "undocumented" status?
This variable is of major importance for the reasons you have outlined in the paper?
"According to the literature, Mexican immigrants, especially those who are undocumented, are less likely to access and use health care services compared to US-born Mexican Americans, other Latinos and non-Latino whites."(p. 2)."According to the literature, Mexican immigrants, especially those who are undocumented, are less likely to access and use health care services compared to US-born Mexican Americans, other Latinos and non-Latino whites."(p. 2).
Table 2: Health related variables stratified by perceived trust in the hosting community. This title is confusing. You say that it is "stratified by "perceived trust in the hosting community", but you do not show the categories of the trust/security variable. Please rethink the title of this table.
Table 3. Ordered logit, OLS, and logistic regressions. This title is also confusing. Why not indicated explicitly in the table which model is OLS, and which is logistic?
Author Response
We appreciate your helpful comments. Here is a point-by-point response.
|
Reviewer 1 |
|
|
Comment |
Response |
|
An interesting topic, but the paper is based on weak data, unfortunately; that is, the sample is non-random and small. Perhaps this study can serve as the groundwork for a more extensive research using scientific sampling with a substantially larger sample. This might be pointed out in the conclusion.
|
We agree with the reviewer that a larger sample would increase statistical power and random selection of participants would increase the generalizability of results to the target population. However, as we discuss in the paper, this a hard-to-reach population and thus having such a sample is unfeasible. While the Consulate mechanisms to schedule appointments might not be fully random, our data collection procedures–in a unique and oftentimes inaccessible site for research– indicate we did not conduct an ad hoc selection of participants. We consider that providing a safe space to answer a sensitive survey outweighs the limitations of having a small sample. Moreover, based on presentations of our data with the consulate’s personnel (not described in the paper), we believe that our data offers key insights on a relevant population. We expect that this publication serves as a research model that we can replicate in more Consulates and States in the US, and thus have larger and more diverse datasets on these issues. Following the reviewer’s suggestion, we point to these strengths and limitations in the Conclusion |
|
1. Surprisingly, a substantively important variable was not stratified in the analysis. Why not stratify on "documented" vs "undocumented" status? This variable is of major importance for the reasons you have outlined in the paper? "According to the literature, Mexican immigrants, especially those who are undocumented, are less likely to access and use health care services compared to US-born Mexican Americans, other Latinos and non-Latino whites."(p. 2)."According to the literature, Mexican immigrants, especially those who are undocumented, are less likely to access and use health care services compared to US-born Mexican Americans, other Latinos and non-Latino whites."(p. 2).
|
Documentation status is a very sensitive information to ask that would potentially negatively affect participation and bias the sample. In this sense, we intentionally did not ask for it. However, the population served by the Consulate is highly undocumented. Considering what the reviewer has raised, having insurance can be a proxy of documentation as to get insurance in the US requires a formal employment with benefits. We expect that the results show this variable was not as important as the frequency by which they receive healthcare when needed.
|
|
Table 2: Health related variables stratified by perceived trust in the hosting community. This title is confusing. You say that it is "stratified by "perceived trust in the hosting community", but you do not show the categories of the trust/security variable. Please rethink the title of this table.
|
Thank you for highlighting this point. We have modified the table based on modifications to our analytic strategy. We corrected the title as follows: “Bivariate associations by self-reported health status”.
|
|
Table 3. Ordered logit, OLS, and logistic regressions. This title is also confusing. Why not indicated explicitly in the table which model is OLS, and which is logistic?
|
We agree with this comment. The analytical strategy changed a lot based on prior comments, but even within the new approach we tried to provide a clearer way of portraying it (see Figures 1 and 3). |
Reviewer 2 Report
Dear authors,
Thank you very much for the opportunity to review this review. I read the work with great interest. However, there are some problems in this paper.
1. The core concept of the research is unclear. The authors regards trust and safety as one concept. However, trust and safety are two different and important concepts, and their connotations are quite different. Although the authors mentioned that the same measurement method was used in literature 33 and 34, but literature 33 measured trust and safety as two concepts, and literature 34 only measured safety.
2. The introduction part of the paper discusses the challenge faced by many Mexican immigrants, which leads to their less use of medical services, thus affecting their health. However, there is no clear discussion about the relationship between trust and health.
3. The research logic of this paper is not very clear. The authors want to discuss the relationship between trust and subjective and objective health. However, in six regression models, the influence of trust on health insurance is constructed. What is the purpose of this regression? Health insurance is a measurement variable under "Healthcare access", while "How frequently has got medical care when needed", another measurement variable under "Healthcare access", serves as an independent variable. What is the relationship between trust and safety, subjective and objective health, and healthcare access? It is not clarified in the paper.
4. Some research conclusions are mainly speculation, such as "Lower levels of trust and safety might also have an indirect path to poor health outcomes by changing patterns of health care access". The authors have not confirmed this conclusion through empirical research.
Based on the above reasons, I think this paper does not meet the requirements for publication
Author Response
We appreciate your helpful comments. Here is a point-by-point response.
|
Reviewer 2 |
|
|
Comment |
Response |
|
1. The core concept of the research is unclear. The authors regards trust and safety as one concept. However, trust and safety are two different and important concepts, and their connotations are quite different. Although the authors mentioned that the same measurement method was used in literature 33 and 34, but literature 33 measured trust and safety as two concepts, and literature 34 only measured safety.
|
The reviewer raised an important topic. Based on this observation, we decided to rearrange the identification strategy so the readers can examine the consistency of the associations along the process. First, we show the models disaggregating the four items. Then we show the distinct association of the strongest effect – neighborhood safety. Afterwards, we show the aggregate measures and test for the interaction. Moreover, in the supplementary material we now show a more detailed run of every variable in the model, first univariate, then bivariate, and ten significant-only associations. The reader will now assess that the main effects remain regardless of the way we estimate the models. We now show a more nuanced story of how these two variables relate with self-rated health. We believe this comment improved the analytic strategy. |
|
2. The introduction part of the paper discusses the challenge faced by many Mexican immigrants, which leads to their less use of medical services, thus affecting their health. However, there is no clear discussion about the relationship between trust and health.
|
The introduction of the paper explains that the reason we are concerned that trust and safety might be important in explaining health is because the ant-migrant rhetoric. Nonetheless, we recognize that healthcare access is influenced by other aspects. Therefore, we removed these models so we can be more specific on its association with health. |
|
3. The research logic of this paper is not very clear. The authors want to discuss the relationship between trust and subjective and objective health. However, in six regression models, the influence of trust on health insurance is constructed. What is the purpose of this regression? Health insurance is a measurement variable under "Healthcare access", while "How frequently has got medical care when needed", another measurement variable under "Healthcare access", serves as an independent variable. What is the relationship between trust and safety, subjective and objective health, and healthcare access? It is not clarified in the paper.
|
We reorganized the analytic strategy to simplify and improve the clarity of our analyses. Based on other comments, we decided to keep subjective health because the survey uses self-reported measures. Moreover, we honed into diverse associations with self-reported health. These allowed us to test different models on our main interest. We dropped the healthcare access models to avoid distractions on our main argument.
|
|
4. Some research conclusions are mainly speculation, such as "Lower levels of trust and safety might also have an indirect path to poor health outcomes by changing patterns of health care access". The authors have not confirmed this conclusion through empirical research.
|
We agree with the reviewer. To avoid speculation, we decided to remove the sentence.
|
Reviewer 3 Report
Overall speaking, the manuscript is well-written and well-structured. However, there are some issues in the manuscript that need the authors to address or clarify.
Conceptualization
•However, I would suggest a clearer conceptual framework should be provided for the study. Preferably, it is a visual or graphical conceptual framework that illustrates how concepts and constructs are linked in the literature and to be investigated in the current study.
•Moreover, there has been a large body of literature on the influence of social capital (including feelings/sense of trust and security) on health status. What are the contributions of the current research to the knowledge construction (e.g. any new theories formulated based on the findings)?
•If possible, some hypotheses should be developed. These hypotheses should be tied with the established or new theories. Specifically, it has been suggested in the literature that social capital affects health through several mechanisms: norms and attitudes that influence health behaviours, psychosocial networks that increase access to health care and psychosocial mechanisms that enhance self-esteem. These mechanisms can form some theoretical bases of the current empirical study.
Research Design
•The researchers relied on the self-rated and self-reported health status coming from a questionnaire survey. However, the title of the paper and abstract mentioned “objective measures of health status”. I cannot see such “objective” measures in the whole research. The number of “diagnosed comorbidities” may not be considered as an objective” measure when this information came from self-reporting of the respondents.
•Actually, it is more interesting to see the differences between immigrants from Mexico and immigrants from other origins. It seems to me that Donald Trump picked particularly on Latinos in his migration policy. The associations between feelings of trust and security and health status may be contingent on the origins of the migrants because they were subject to different levels of discrimination.
Implications
•The current manuscript looks rather fact-finding. What specific policy implications can be drawn upon the findings of the research?
Author Response
We appreciate your helpful comments. Here is a point-by-point response.
|
Reviewer 3 |
|
|
Comment |
Response |
|
I would suggest a clearer conceptual framework should be provided for the study. Preferably, it is a visual or graphical conceptual framework that illustrates how concepts and constructs are linked in the literature and to be investigated in the current study.
|
Agreed. While we did not provide a visual/graphical framework, we provided a clearer explanation, which partially emerges from a more focused analysis on a single dependent variable.
|
|
Moreover, there has been a large body of literature on the influence of social capital (including feelings/sense of trust and security) on health status. What are the contributions of the current research to the knowledge construction (e.g. any new theories formulated based on the findings)? |
As stated in the text, we believe that the three main contributions are the type of population and data collection location; the timing (i.e. highly anti-immigrant context); as well as the disaggregation of the effects of the items and its comparison with scores.
|
|
If possible, some hypotheses should be developed. These hypotheses should be tied with the established or new theories. Specifically, it has been suggested in the literature that social capital affects health through several mechanisms: norms and attitudes that influence health behaviours, psychosocial networks that increase access to health care and psychosocial mechanisms that enhance self-esteem. These mechanisms can form some theoretical bases of the current empirical study. |
Agreed. Thanks for this comment, we added some of these discussions to the introduction section.
|
|
Research Design
•The researchers relied on the self-rated and self-reported health status coming from a questionnaire survey. However, the title of the paper and abstract mentioned “objective measures of health status”. I cannot see such “objective” measures in the whole research. The number of “diagnosed comorbidities” may not be considered as an objective” measure when this information came from self-reporting of the respondents.
|
We agree with this observation. All measures are self-reported, and, in that sense, they are all subjective. We decided to focus on subjective health to test our main hypothesis, which it is now stated at the end of the Introduction. We changed the title to avoid any confusion on this regard. |
|
•Actually, it is more interesting to see the differences between immigrants from Mexico and immigrants from other origins. It seems to me that Donald Trump picked particularly on Latinos in his migration policy. The associations between feelings of trust and security and health status may be contingent on the origins of the migrants because they were subject to different levels of discrimination.
|
Unfortunately, the data collected at the Mexican Consulates makes it impossible to make comparisons with other nationalities. We agree that this is an interesting hypothesis, but we are unable to test it with these data. |
|
Implications
•The current manuscript looks rather fact-finding. What specific policy implications can be drawn upon the findings of the research?
|
Thanks for highlighting this. We added a full paragraph in the discussion section addressing the implications. |
Round 2
Reviewer 2 Report
The authors have made substantive amendments to the paper. The research issues and methodsof the revised version are clearer. However, the following two issues still need to be further improved.
1. The “INTRODUCTION”section mainly discusses the relationship between immigration policy and access to health care services, and the relevant literature on the relationship between trust, safety and subjective health should be strengthened.
2. "95% confidence intervals in brackets" was not displayed in Figure 3.
Author Response
We appreciate the comments provided, which will indeed improve the manuscript.
Comment 1. The “INTRODUCTION”section mainly discusses the relationship between immigration policy and access to health care services, and the relevant literature on the relationship between trust, safety and subjective health should be strengthened.
The following paragraph was added to the introduction:
"In addition, evidence shows that changes in immigration enforcement actions have resulted in a greater fear of deportation (even among naturalized citizens and legal permanent residents), thus generating an atmosphere of anxiety, stress, and depression that may ultimately result in the worsening of health outcomes and well-being of immigrant families (23-25). Prior work documents that a large proportion of Hispanics (both citizens and noncitizens) feel unsafe living in the U.S. because of the anti-immigrant narratives, which has fostered fear and mistrust (26). This is likely to affect the perceptions about the communities and contexts were immigrants live (27), including aspects such as neighborhood safety and trust. This is worrisome as prior research suggests that such neighborhood and community perceptions can have implications in subjective (28-30) and objectives measures of health (31).
2. "95% confidence intervals in brackets" was not displayed in Figure 3.
This was an error in the footnote of Figure 3. The Figure footnote has been amended to allow the audience to have a clear understading of what it is portraying.